Improved angelization technique against background knowledge attack for 1:M microdata

Fazal Rabeeha 1
Khan Razaullah 2
Anjum Adeel 3
Syed Madiha Haider madiha@qau.edu.pk 3
Khan Abid 4
Rehman Semeen semeen.rehman@tuwien.ac.at 5
1 Department of Computer Science, COMSATS Institute of Information Technology , Islamabad , Pakistan
2 Department of Computer Science, University of Engineering and Technology , Mardan , Pakistan
3 Institute of Information Technology, Quaid-i-Azam University , Islamabad , Pakistan
4 College of Science and Engineer, University of Derby , Derby , United Kingdom
5 Institute of Computer Technology, Technische Universität Wien , Wien , Austria
Maglaras Leandros
Electronic publication date: 2023 Mar 15
Publication date: 2023
Volume: 9
Electronic Location ID: e1255
Received 2022 Apr 19; Accepted 2023 Jan 24
Copyright: ©2023 Fazal et al.
Copyright year: 2023
Copyright holder: Fazal et al.
License: This is an open access article distributed under the terms of the Creative Commons Attribution License, which permits unrestricted use, distribution, reproduction and adaptation in any medium and for any purpose provided that it is properly attributed. For attribution, the original author(s), title, publication source (PeerJ Computer Science) and either DOI or URL of the article must be cited.
License URL: https://creativecommons.org/licenses/by/4.0/

Keywords: Security, Privacy, Internet of Things (IoT), Anonymity

Funding: TU Wien Bibliothek through its Open Access Funding Programme This work was supported by TU Wien Bibliothek through its Open Access Funding Programme. The funders had no role in study design, data collection and analysis, decision to publish, or preparation of the manuscript.

==============================
With the advent of modern information systems, sharing Electronic Health Records (EHRs) with different organizations for better medical treatment, and analysis is beneficial for both academic as well as for business development. However, an individual’s personal privacy is a big concern because of the trust issue across organizations. At the same time, the utility of the shared data that is required for its favorable use is also important. Studies show that plenty of conventional work is available where an individual has only one record in a dataset (1:1 dataset), which is not the case in many applications. In a more realistic form, an individual may have more than one record in a dataset (1:M). In this article, we highlight the high utility loss and inapplicability for the 1:M dataset of the θ-Sensitive k-Anonymity privacy model. The high utility loss and low data privacy of (p, l)-angelization, and (k, l)-diversity for the 1:M dataset. As a mitigation solution, we propose an improved (θ∗, k)-utility algorithm to preserve enhanced privacy and utility of the anonymized 1:M dataset. Experiments on the real-world dataset reveal that the proposed approach outperforms its counterpart, in terms of utility and privacy for the 1:M dataset.

Introduction

Modern technologies, such as the Internet of Things (IoT) and Big Data are the key enablers to revolutionizing today’s modern society in different fields, for example, the Electronic Health Records (EHRs) (Dang et al., 2019; Muftuoglu, Kızrak & Yildirim, 2022; Amin et al., 2022). The government or private organizations collect the EHRs via the IoT devices and share them for further statistical analysis and policymaking (Moonsamy & Singh, 2022; Sheikhtaheri et al., 2022). However, the EHRs belong to an individual, these are very confidential and crucial in the medical information control system. Sharing such data without privacy implementation is unlawful, because of the possibility of privacy breach and misuse of data (Dang et al., 2019; Sun et al., 2018; Al-Khafajiy et al., 2019; Al-Khafajiy et al., 2018; Fazal et al., 2022), as shown in Fig. 1, where an attacker compromises the privacy of an individual. Almost all previous anonymization techniques deal with the classical type of data where one person has only one record, i.e., 1:1 dataset; this may not be applicable for all tasks, e.g., health complicated analysis (Jayapradha et al., 2022), etc. In a real-world scenario, an individual may have multiple health data records in a dataset or have multiple datasets, known as 1:M microdata (Gong et al., 2017).

Figure 1 An example of attacker model.

This article studies the state-of-the-art θ-Sensitive k-Anonymity (Khan et al., 2020a), (p, l)-angelization (Kanwal et al., 2019), and (k, l)-diversity (Gong et al., 2017) algorithms, to highlight respectively their inapplicability for 1:M microdata, privacy breach, and the low data utility. The θ-Sensitive k-Anonymity (Khan et al., 2020a) is a numerical measure of privacy strength. The θ denoted the threshold for achieving diversity for sensitive attributes. The θ threshold value is used to achieve diversity through the variance in a group of records (i.e., Equivalence Class - EC). It anonymizes the 1:1 type of data, however, is not applicable for privacy implementation in 1:M microdata, because of the variance calculation inside an EC where each disease value belongs to a specific individual record. If there are more than one disease values (i.e., 1:M data) then the proposed variance calculation method of θ-Sensitive k-Anonymity (Khan et al., 2020a) fails. Also, θ-Sensitive k-Anonymity (Khan et al., 2020a) algorithm has not proposed any suitable technique for improving the utility of the data.

In the (k, l)-diversity, k is used to protect quasi identifiers (e.g., age, gender, zipcode), and l for the sensitive attributes (e.g., disease, salary) in an EC, so (k, l)-diversity is used to protect sensitive and quasi attributes. But (k, l)-diversity (Gong et al., 2017) partial generalizes the SA values, which causes the sensitive vertical attack (sVer) for the attacker to breach the privacy of data (see Scenario I and Definition 5 for detail). Another privacy focused approach; is the (p, l)-angelization, here the (p, l) shows the extension of the model of p-sensitive k-anonymity (Ye, Yang and Liu, Yu and Wang, Chi and Lv, Dapeng and Feng, Jianhua, 2009) and the angelization is used to splits tables into the quasi and sensitive table. The(p, l)-angelization (Kanwal et al., 2019) for 1:M-MSA data (i.e., an individual having multiple records and multiple sensitive or confidential attributes), has very low utility considerations. Although, the(p, l)-angelization (Kanwal et al., 2019) creates two tables linking through a bucket id (BID). However, that linkage is useless and has no contribution in utility improvement because of the one-to-one correspondence between the two tables. The initial work for 1:M in (k, l)-diversity partially generalizes the SA, giving a privacy leakage window to the adversary and revealing the current medical status of a person with the help of some background knowledge.

In this article, we propose (θ∗ , k)-utility technique for 1:M microdata. The (θ∗ , k)-utility algorithm overcomes the limitations of (k, l)-diversity, θ-Sensitive k-Anonymity, and (p, l)-angelization by improving data utility and its applicability for 1:M microdata (Gong et al., 2017; Khan et al., 2020a; Kanwal et al., 2019). The motivation subsection discusses in detail the limitations in (k, l)-diversity, θ-Sensitive k-Anonymity and in (p, l)-angelization approaches (Gong et al., 2017; Khan et al., 2020a; Kanwal et al., 2019).

Motivation

Most of the published work in data privacy focuses on either privacy or utility. Keeping a balance between them has always remained an open research problem among researchers. Therefore, the main focus of this research is to come up with an optimized solution where the privacy of the sensitive attributes does not affect the utility of the quasi attribute. A novel solution is needed to independently publish the sensitive table and quasi data without affecting the utility and privacy of the anonymized data. Also, the data should be real-life realistic data, i.e., 1:M data.

The Table 1 of EHRs having 1:M microdata of different individuals are classified into; unique identifier attributes—ID (e.g., name, national identification number, passport number), quasi identifier attributes—QI (e.g., age, gender, zip code, country, religion), and confidential or sensitive attributes—SA (e.g., Symptoms).

Table 1 Original microdata table T.

Patient Record ID	Tuple ID	Name	Age	Zipcode	Gender	Symptoms	
p1	t1	Susan	40	2139	Female	Flu	
	t2	Susan	40	2139	Female	Chills	
	t3	Susan	40	2139	Female	Fever	
p2	t4	Ronald	45	2545	Male	Difficult Breathing	
	t5	Ronald	45	2545	Male	Lungs Infection	
p3	t6	Keran	45	2238	Female	Cough	
	t7	Keran	45	2238	Female	Chest Pain	
p4	t8	Heather	38	2843	Male	Stomach Pain	
	t9	Heather	38	2843	Male	Lungs Infection	
p5	t10	Cytnthia	42	2341	Female	Headache	
	t11	Cytnthia	42	2341	Female	Tiredness	
	t12	Cytnthia	42	2341	Female	Flu	
p6	t13	Peter	40	2548	Male	Headache	
	t14	Peter	40	2548	Male	Flu	
p7	t15	Helan	45	2544	Female	No symptoms	
p8	t16	Jonas	32	2538	Male	Headache	

Before publishing the data, removing only the ID is not enough because the attacker (i.e., intruder) that can re-identify individual record respondents using some background knowledge—BK (i.e., external source of knowledge that helps in re-identification of an individual) by combining the certain pattern of QIs with some externally available data (e.g., census or voting data), to perform the linking attack, (see Fig. 1).

The θ-Sensitive k-Anonymity is a state-of-the-art privacy algorithm, which is applicable to preserve privacy for a single sensitive (SA). However, in addition to low data utility, it cannot work directly for any other type of data. Similarly, the (p, l)-angelization (Kanwal et al., 2019), does not focus for data utility improvement. The following scenario explains θ-Sensitive k-Anonymity approach inapplicability for 1:M microdata and its low data utility, and also (p, l)-angelization (Kanwal et al., 2019) high utility loss.

• Scenario I Sensitive Vertical Attack (sVer):

The initial work for 1:M in (k, l)-diversity (Gong et al., 2017) partially generalize the SA, which give a privacy leakage window to the adversary and reveal the current medical status of a person with the help of some background knowledge; information about a most recent visit or the first visit (either his disease is progressing (worsening), or recovering).

The (k, l)-diversity used a transaction generalization technique (Gong et al., 2017) to apply k-anonymity on SAs. The lowest common cut on transaction generalization hierarchy, leaves the single SA from different sub-set unprotected and vulnerable, as shown in Fig. 2. In Fig. 2, if a patient’s SAs of multiple records are headache, flu, tiredness, and chills, the (k, l)-diversity will make a lowest common cut on transaction generalization hierarchy and generalize the SAs into ‘Low, Chills’. The Low node covers headache, flu, tiredness leaf-nodes and Chills cover chills; which remains ‘unprotected and vulnerable and also keeps the Cough leaf-node isolated, which any potential adversary can exploit both the Chills and Cough leaf-nodes, this causes sVer attack, due to partially generalized sensitive attributes.

Figure 2 Sensitive attributes hierarchical structure.

• Scenario II inapplicability for 1:M microdata:

The θ-Sensitive k-Anonymity algorithm (Khan et al., 2020a) is based on the θ-threshold value; which is a multiplied component of variance, and observation1. The variance calculation in each EC (i.e., group of anonymous records) is concerning with the frequency distribution of SA values.

If there are more than one SA values for a single record (1:M microdata), then the variance calculation in an EC is not possible. For example to check the inapplicability of θ-Sensitive k-Anonymity for 1:M microdata, consider Table 1 having 1:M microdata. For an anonymized EC, the θ is obtained as Eq. (2), based on variance is obtained as Eq. (1). That shows inapplicability of the threshold value for the 1:M dataset. (1) σ2=ΣFX2N−ΣFXN2

(2) θ=VarianceofafullydiverseECσ2×Observation1μ.

It is obvious that the sensitive values and the corresponding frequency distributions can not be applied to 1:M microdata in Table 1. The reason is that the Symptoms attribute in Table 1 has more than one sensitive value for a specific individual record, which does not represent a frequency distribution for a specific sensitive value. Similarly, if Table 1 is transformed to Table 2; having a complete record in a single tuple. Again, the SA values are more than one in a single tuple, and the θ-Sensitive k-Anonymity approach can not be directly applied to implement diversity in an EC. So, the variance-based θ threshold calculation to obtain a diverse EC is not applicable for 1:M microdata.

Table 2 θ-Sensitive k-Anonymity inapplicability for 1:M microdata.

Patient record ID	Name	Age	Zipcode	Gender	Symptoms	
p1	Susan	40	2139	Female	<Flu, Chills, Fever >	
p2	Ronald	45	2545	Male	<Difficult Breathing, Lungs Infection >	
p3	Keran	45	2238	Female	<Chest pain, Cough >	
p4	Heather	38	2843	Male	<Stomach pain, Lungs Infection >	
p5	Cytnthia	42	2341	Female	<Flu, Tiredness, Headache >	
p6	Peter	40	2548	Male	<Flu, Headache >	
p7	Helan	45	2544	Female	<No Symptoms >	
p8	Jonas	32	2538	Male	<Headache >	

• Scenario III High utility loss:

It is based on two approaches.

(a) The θ-Sensitive k-Anonymity (Khan et al., 2020a) approach only focuses the attribute disclosure prevention and does not have any consideration for improving the utility of the anonymized data. The algorithm for the θ-Sensitive k-Anonymity begins with checking the k for its minimum value and no further utility improvement consideration is available in the rest of the algorithm.

(b) The (p, l)-angelization (Kanwal et al., 2019) splits the microdata table T into Quasi Table (QT) and Sensitive Table (ST). However, the sensitive buckets inside each table have a one-to-one correspondence with one another, which affects the utility in the QT.

Contribution

The main contributions of the proposed (θ∗, k)-utility privacy algorithm are as follows.

• The proposed (θ∗, k)-utility privacy algorithm, categorizes the SA values of 1:M microdata into Low, Mild, High, Severe, and A-symptomatic values, based on the category Table 3 to reshape the original microdata Table 1 into Table 4, for the purpose to get the 1:1 microdata. The SA 1:M record values are replaced with category table SA values. If the SA values are repeated in more than one category, the higher category value is considered and ignored the lower one and stored in history table.

• The proposed algorithm using the angelization approach, anonymizes the microdata T in Table 1 into QT and ST (see Section 5) and are linked through the Bucket ID (BID) using the one-to-many correspondence (i.e., QS-Loose Linkability) for improving utility and privacy, instead of one-to-one correspondence.

• Based of the above points, the experiment results demonstrate the out performance of the proposed (θ∗, k)-utility privacy algorithm, as compared to its counterparts in terms of utility and privacy.

The rest of the article is organized as follows. Section 2 discusses Related Work. Section 3 covers the Preliminaries, and Section 4 discusses the HLPN analysis of previous models. Section 5 discusses the proposed algorithm. Section 6 discusses the Experimental Analysis. Section 7 depicts some Discussion on the base and current proposed work. Section 8 concludes the article with possible future research directions.

Table 3 Category table - CtgT.

ID	Category	Symptoms	
1	A-symptomatic	No Symptoms	
2	Low	Flu, Headache, Tiredness	
3	Mild	Loss of Appetite, Cough, Chills, Fever	
4	High	Stomach Pain, Difficult Breathing, Chest Pain	
5	Severe	Lung’s Infection, Respiratory Problem	

Table 4 Original microdata with categorical sensitive values.

Patient record ID	Tuple ID	Name	Age	Zipcode	Gender	Symptoms	
p1	t1	Susan	40	2139	Female	Low	
	t2	Susan	40	2139	Female	Mild	
	t3	Susan	40	2139	Female	Mild	
p2	t4	Ronald	45	2545	Male	High	
	t5	Ronald	45	2545	Male	Severe	
p3	t6	Keran	45	2238	Female	Mild	
	t7	Keran	45	2238	Female	High	
p4	t8	Heather	38	2843	Male	High	
	t9	Heather	38	2843	Male	Severe	
p5	t10	Cytnthia	42	2341	Female	Low	
	t11	Cytnthia	42	2341	Female	Low	
	t12	Cytnthia	42	2341	Female	Low	
p6	t13	Peter	40	2548	Male	Low	
	t14	Peter	40	2548	Male	Low	
p7	t15	Helan	45	2544	Female	A-symptomatic	
p8	t16	Jonas	32	2538	Male	Low	

Related Work

We studied and analyzed different existing methods and approaches in the literature. Individual privacy is guaranteed with anonymized data since encryption cannot be used for publicly available data to preserve data privacy (Shahzad et al., 2018; Michalas, 2019). The consensus is before publishing data anonymize it to achieve data privacy. The reason is that, with the anonymized data an individual privacy is preserved. We organize the anonymize techniques into 1:1 and 1:M microdata.

Privacy for 1:1 microdata

Lv & Piccialli (2021) approach is based on the combination of k-anonymity and algorithm of K-A-DP to preserve data privacy. It reduces risks of privacy loss, but It only focuses on numeric values. The author discussed that DP is used for protection when stored in a different place and applied to stored data. It also provides data privacy of electronic health records that cause utility loss (Choudhury et al., 2019). Tu et al. (2018) discussed the model used for preserving the vulnerability of records using the k anonymity. It is used to minimize vulnerability to identify the records, but identifying sensitive values can not completely protect that cause breach. Liu & Li (2018) proposed an approach that is k-anonymous based on clustering. This process is time-consuming in terms of finding anonymous equivalence.

The article (Nasir et al., 2017) prevents the attribute disclosure with skewness attack, which extended the distribution scheme based on the weighted table. It provides low data privacy for the same sensitive values due to non-generalized quasi values. Lee & Lee (2017) proposed a model that used the identification factors to predict the re-identification of quasi attributes. The identification probability is based on some factors. But also, it can not fully minimize the aspect of re-identification. Majeed, Ullah & Lee (2017) proposed the protection of personal identity information from vulnerability. It provides data privacy to minimize vulnerable records but still has low diversity. Yaseen et al., (2018) proposed a model based on conventional, divisor, and cardinality hierarchy. That generates generalization hierarchies but does not focus on textual values. Anjum et al. (2018) extended the p+-sensitive k-anonymity and improved this in a balanced p+-sensitive k-anonymity model. It has low diversity for sensitive attributes.

Raju, Seetaramanath & Rao (2019) proposed a model based on slicing, which correlated quasi attributes. The suppression of sensitive attributes depends on the threshold used to create one sensitive value used for all sensitive attributes. A lot of QI and SA suppression may cause huge utility losses. Song et al. (2019) proposed a model that used k-anonymous data using noise addition and randomization for categorical data. It may be used for privacy but not use for long-range numeric data. The author presents the improved k-Anonymity with l diversity; the k anonymity and l diversity protect the identity disclosure. It provides data utility (Jain, Gyanchandani & Khare, 2020). The θ-Sensitive k-Anonymity is a variance-based θ threshold calculation to obtain diverse Equal Classes. Although the θ-Sensitive k-Anonymity prevents attribute disclosure with sensitive variance and similarity, it cannot be used directly for 1:M microdata (Khan et al., 2020a).

Privacy for 1:M microdata

The 1:M type of data is a more realistic form of records stored in EHRs. The(k, l)-diversity model based on 1:M generalization, which prevented attribute disclosure (Gong et al., 2017). But it provides low utility and privacy for data. Wang et al. (2019b) the algorithm is based on clustering, various decision functions used, but is vulnerable to sensitive attributes. It provides utility for quasi attributes but still has vulnerable to sensitive attributes. The author discussed the approach based on providing privacy for vulnerability disclosure using the model of 1: M MSA- (p, l)-diversity. The attribute disclosure is prevented through this model but also has low data utility due to one-to-one linkage (Kanwal et al., 2021). Anjum et al. (2018) proposed a heuristic approach to protect the sensitive and quasi values using splitting. It provides privacy but gives low data utility due to 1:1 correspondence. The l-anatomy focuses on the utility of data (Anjum et al., 2019). But they publish the sensitive attributes without generalization. This approach deals with achieving utility, but that may not provide enough security. The generic 1:M data privacy (G-model) model uses the signature method to achieve privacy (Albulayhi, Tošić & Sheldon, 2020). However, it has a limitation in which attackers can attack using the signature values to spot records.

The author focuses on COVID-patient data, where the privacy keeps through spatial k-anonymity. But it has limitations to achieving privacy, causing loss of privacy (Iyer et al., 2021). This author discussed the approach for privacy-preserving using the k-Anonymity. The focus is using the values of (p, and l) as a threshold. But it cannot prevent the sensitive variance attack, although this approach uses k-anonymity to improve the privacy. It provides low data privacy (Zhang et al., 2017). The author used the approach of (α, k) to protect privacy. The Poker dataset is used to measure results, and it cannot protect from the attack to sensitive attributes. It protects from identity disclosure. The attacker accesses the data using a background knowledge attack with low data utility (Wang et al., 2019a). The (p, l)-angelization (Kanwal et al., 2019) for 1:M-MSA data has very low utility considerations. Although, (p, l)-angelization creates two tables linking through a bucket id (BID). However, that linkage is useless and contributes to utility improvement because of the one-to-one correspondence between the two tables.

The previous approaches do not provide privacy for sensitive attributes and data utility. If someone deals with they cannot directly be used for the 1:M dataset or cannot prevent from background knowledge attack on sensitive attributes for 1:M COVID-19 data (Gong et al., 2017; Kanwal et al., 2019); so to preserve the 1:M dataset privacy and data utility, we overcame these limitations and extended the θ-Sensitive k-Anonymity because the θ-Sensitive k-Anonymity deals with similarity and variance attack for 1:1 and overcome the limitation of privacy and utility in 1:M microdata to preserve the COVID-19 patient data privacy.

Preliminaries

Let the input table T = {ID, QI, SA} (i.e., Table 1), having 1:M microdata. The ti ϵ T is a tuple that represents an individual i having complete or partial record details, depends on the number of tuples that belongs to an individual i. The ti is a component combination of ID ={ A1id,A2id, A3id,⋯, Anid}, QI ={A1qi,A2qi,A3qi,⋯,Anqi}, and SA = {A1sa,A2sa,A3sa,⋯,Ansa}. In the anonymized form the Aids are removed, and a suitable anonymization technique is applied on Aqi and Asa, to prevent the identity and attribute disclosures. However, such anonymization techniques should be strong enough to prevent any possible attack; e.g., membership or non-membership attack (ma or nma), sensitive variance attack (sva), categorical similarity attack (csa), sensitive vertical attack (sVer) or any other background knowledge attack (bka). The Table 5 summarizes all the notations used in this article.

Table 5 Summary of notations.

Symbols	Description	
T	1:M Microdata Table	
k	k-anonymity	
EC	Equivalence Class	
 Γ	T populated with CtgT	
ECs	Equivalence Classes	
tisa	Generalized Sensitive Attributes	
A id	Explicit Attributes	
BK	Background Knowledge	
A qi	Quasi Attributes	
A sa	Sensitive Attributes	
Dnsa	Set of distinct sensitive attributes	
disa	Distinct SA values in dataset	
τ	Individual tuple from transform Table T	
ℏ	Individual history tuple from history table ℍ	
sbt	Sensitive buckets	
QT	QT contains all QI attributes with QID	
ST	ST consists of (Bucket-ID, SAs)	
sVer	Sensitive Vertical Attack	
CtgTcat	CtgT generalized SA	
tisa	Sensitive attribute tuples from 1:M	
r i	Single record having may rows	
genData	QT	
t i	Tuples from 1:M belongs to single individuals	
ma	Membership Attack	
nma	Non-membership Attack	
sva	Sensitive Variance Attack	
csa	Categorical Similarity Attack	
QS-Loose Linkability	QT & ST linked one-to-many correspondence	

The speculation is an adversary, which can access information of any individual using some background knowledge. The adversary model is represented as below.

QT = {A1qi,A2qi,A3qi,⋯,Anqi}, QT contains all QI attributes and Bucket-ID (BID).

ST = { A1sa,A2sa,A3sa,⋯,Ansa}, ST consists of the SAs and Bucket-ID (BID), that is linked to the BID in QT.

BID: An identifier between the QT and ST, which links the buckets in both tables through the one-to-many correspondence, known as QS-loose linkability.

BK = {QT, ST, BID, any publicly available information}, where BK is the background knowledge of an adversary.

Membership attack (ma): The privacy breach due to the identification of a particular sensitive value that belongs to an individual i can be linked with a specific group of QI attributes due to membership knowledge (mk) is called a membership attack.

Non-membership attack (nma): The privacy breach due to the identification of a particular sensitive value that belongs to an individual i can not be linked with a specific group of QI attributes due to non-membership knowledge (nmk) is called a non-membership attack.

Sensitive variance attack (sva): The low variability of SA values in an EC from different SA categories in a category table is called a sensitive variance attack.

Categorical similarity attack (csa): The SAs in an EC is obtained from a single category of the category table, which may narrow the adversary’s knowledge to attack, called categorical similarity attack.

Sensitive vertical attack (sVer): The correlation and generalization of SA values vertically from different hierarchical levels, to isolate a SA value for re-identification of an individual, with the help of background knowledge called sensitive vertical attack.

QS-loose linkability: The proposed QT and ST are loosely (i.e., independently) linked through one-to-many correspondence, for improved privacy and utility, instead of one-to-one tight correspondence is called loose likability.

Transformation (Gong et al., 2017): The records of same individuals have the same QID values in the dataset. The dataset can be transformed by merging all the same individual’s QID values to a single set. In the transformed dataset each individual has only one record consisting of his/her QID and SAs.

Angelization (Kanwal et al., 2019; Xiao & Tao, 2006): The sensitive partitioning A = {A1, A2, ⋯, An}, and the quasi partitioning B = {B1, B2, ⋯, Bm} of the microdata Table T, an Angelization of Table T produces two tables: ST and QT, such that, ST consists of Bucket-ID and SAs, where SAs represents the Sensitive Attribute column of Table T. QT contains all QI attributes with QID belonging to Table T.

High-Level Petri Nets (HLPN) (Malik, Khan & Srinivasan, 2013): A graphical and mathematical representation for examining the information control. It consists of 7-tuples; N = (P, T, F, ϕ, Rn, L, M0). The static semantics are shown using L, ϕ and Rn whereas F, P, and T provide the dynamic structure. The P is the set of all places, where a single place is represented by a cycle. T is the set of all transitions (i.e., rectangular boxes in HLPN), where transitions show the changes encounter in the system. The relation between P and T is such that P ∩ T = ϕ, P ∪ T ≠ ϕ. The rules for these transitions are represented by Rn. F represents the information flow such that F ⊆ (P × T) ∪ (T ∪ F). The data types are mapped to the places P through ϕ, L refers to a label on F, and M0 represents the initial marking.

HLPN analysis of previous models

The formal modeling of (k, l)-diversity and θ-Sensitive k-Anonymity are performed here to reveal the way how an adversary can perform an attack.

(k, l)-diversity

The formal modeling and analysis reveals the way how an adversary can perform a sensitive vertical attack on the 1:M dataset as shown in Fig. 3. The working of (k, l)-diversity given in Kanwal et al. (2019) from Rule (1) to (12), where data is taken from data owner and data publisher anonymized it. The types are shown in Table 6 and data placed with description in Table 7. The black rectangular boxes show transition arrows showing the flow, and circles represent places or sub-part of the system. The data owner, data publisher, and the adversary are the entities.

Figure 3 HLPN for (k, l)-diversity.

Table 6 Types used in HLPN for (k, l)-diversity.

Symbols	Description	
PID	Patient ID	
TID	Define level of k-anonymity	
Contf	True or False	
QI	Qausi idetifier	
Int	Define QI using integer	
Chr	Define QI using char	
GSA	Generalized Sensitive Attribute	
TSA	Transformed sensitive attribute	
PIDqi	Reveal identity of patient using qausi attributes.	

Table 7 Mapping of data types in HLPN for (k, l)-diversity.

Symbols	Description	
ϕ(MDT)	ℙ (QI ×SA ×PID )	
ϕ(TMDT)	ℙ (QI ×SA ×TID )	
ϕ(Flag-tf)	ℙ (Condtf)	
ϕ(G-list)	ℙ (QI ×SA ×TID )	
ϕ(k)	ℙ(k)	
ϕ(L)	ℙ(l)	
ϕ(SP-node )	ℙ (GSA)	
ϕ(Sub-p )	ℙ (QI ×GSA ×TID )	
ϕ (IT-v)	ℙ (QI ×ITSA ×TID )	
ϕ(CT-v)	ℙ (QI ×ITSA ×TID )	
ϕ(Dim-v)	ℙ (Int× chr)	
ϕ(D-type )	ℙ (Int× chr)	
ϕ(Thres hold)	ℙ (Thr-v)	
ϕ(Published Data)	ℙ (QI × SA )	
ϕ(BK)	ℙ (PID ×QI ×SA )	
ϕ(Sa disc)	ℙ (QI ×SA ×PID )	

The first transition shows input taken from the data owner of 1:M data after taking, anonymizing the data, and publishing it. After publishing that data adversary can perform an attack on it. The anonymization process starts from transformation and then checks the individual record, distinct at least k-1. After this in each subpartition, the GSA record is balanced and after that stored in IT-v. The G-list consists of records less than k. The CT has contains records of IT-v and G-list. Then the category of numeric and categorical is checked by choose attribute and check dim, after this update values according to the threshold. Finally, Mndrn is used for generalization and checking the splits of SA subpartition, if it can not then places the records into G-list.

The sVer attack performed on (k, l)-diversity because of partially generalization of SA values. It can be correlated with any generalized categorical SA value. Since all the generalized categorical SA values are on a different level in the hierarchy, SA values on a different level can be vertically correlated to breach the privacy of an individual in Rule 3 as. (3) R(sV er):=∀i47ϵ,x47,∀i48ϵx48,∀i49ϵx49|sVer−atki472,i482→i491=i22∧i492=i23.

In Rule 3 an attacker can be attacked due to the lowest common cut on transaction generalization hierarchy, leaving the single SA from different sub-set unprotected and vulnerable which leads toward correlation of the background information with published data that ultimately caused an attack on sensitive attributes known as sVer.

θ-Sensitive k-Anonymity

The formal modeling analysis reveals the way how an adversary can perform a sensitive variance attack on 1:M data because of inapplicability of θ-Sensitive k-Anonymity for 1:M microdata.

The working of theta given in Khan et al. (2020a) from Rule (1) to (8), where data is taken from data owner and data publisher anonymized it as shown in Fig. 4. The types are shown in Table 8 and data placed with description in Table 9. The black rectangular boxes show transition arrows showing the flow, and circles represent places or sub-part of the system. The data owner, data publisher, and the adversary are the entities.

Figure 4 HLPN for θ-Sensitive k-Anonymity.

Table 8 Types used in HLPN for θ-Sensitive k-Anonymity.

Symbols	Description	
M	EC size	
Condition	True or False	
σ	The notation of sigma	
μ	The notation of observation value	
θ	The notation of threshold value	
fnd ECb	Found the EC of b	
Adjust ECc	The EC of c adjust	
Adjust ECn	The EC of n adjust	
Var ECS	Different ECS varaince.	
V-ad ECn	The variance of EC adjust for class n	
V-adjust ECc	The variance of EC adjust for class c	

Table 9 Mapping of data types in HLPN for θ-Sensitive k-Anonymity.

Symbols	Description	
ϕ(1:MT)	ℙ (Aid× Aqi× Asa	
ϕ(MMT)	ℙ (ECC× ECb× ECn×k)	
ϕ(k)	ℙ (k)	
ϕ(cond)	ℙ(Condition)	
ϕ(sigma)	ℙ (σ)	
ϕ(mu)	ℙ (μ)	
ϕ(theta)	ℙ (θ)	
ϕ(Fnd ECb)	ℙ (ECb)	
ϕ(Var ECS)	ℙ (VECc× VECb× VECn)	
ϕ(Adjust ECc)	ℙ(ECc)	
ϕ(Adjust ECn)	ℙ (ECn)	
ϕ(ST ECn−1	ℙ (ECn−1)	
ϕV-adjust ECc	ℙ (V ECc)	
ϕV-adjust ECn	ℙ (V ECn)	
ϕNead ns	ℙ (VECc× Aid× Aqi× Asa)	
ϕP data	ℙ ( Aqi× Asa)	
ϕBK	ℙ ( Aid× Aqi)	
ϕSa disc	ℙ (Aiid× Aisa× Aiqi)	

The first transition shows input is taken from the data owner of 1:M data after taking, anonymizing the data, and publishing. After publishing that data adversary can perform an attack on it. The anonymization process starts from taking input and checking k value. After it calculates threshold using var() function, and if needed variance value adjust using adj(), swap using swap() or adding noise using Add ns(). For 1:M dataset, SA values are more than one in a single tuple in that way θ-Sensitive k-Anonymity approach can not be directly applied to implement diversity in an EC. Because the variance-based θ threshold calculation to obtain a diverse EC is not applicable for 1:M microdata. So a sensitive variance attack can be performed on the published data using any background knowledge. In Rule 4 as: (4) R(SV A):=∀i38ϵ,x38,∀i40ϵx40,∀i41ϵx41,∀i2ϵx2|sva−atti382,i402→i431=i21∧i412=i23

The EC produced, can not prevent the sva in definition 3, and csa in definition 4, because of inappliabilty of θ-Sensitive k-Anonymity for 1:M dataset, therefore the attack can be performed on 1:M dataset.

Proposed (θ∗, k)-utility

The purpose of anonymization should not be singleton to either privacy or utility. And at the same time the proposed approach must be strong enough to prevent any possible attack and also provide quality of data. Therefore, the purposed (θ∗, k)-utility algorithm in this article not only anonymizes 1:M data to prevent possible attacks, e.g., ma, nma, sva, csa, and sVer, but also provide improved quality data.

In our proposed approach we apply full generalization of SA, that can bee seen in the transformation of Table 4 into Table 10. Further, we apply partitioning of the QI and SA, and use QS-loose linkability for one-to-many correspondence to prevent any possible privacy leakage. To prevent the sVer attack, identified in (k, l)-diversity (Gong et al., 2017), the SA are placed in ECs using the θ-Sensitive k-Anonymity (Khan et al., 2020a) approach. However, since the anonymized data is 1:M, and the θ-Sensitive k-Anonymity (Khan et al., 2020a) cannot be directly applied because of the variance calculation for each EC. Therefore, the leaf-nodes of SA in Fig. 2 are generalized based on CtgT Table 3 and applying the θ-Sensitive k-Anonymity (Khan et al., 2020) to implement required diversity in each EC. The θ-Sensitive k-Anonymity (Khan et al., 2020) approach is a simple numerical measure of privacy strength which ensures a strong privacy implementation for each EC and hence for the complete dataset.

Table 10 Transformed Microdata T.

Patient Record ID	Age	Zipcode	Gender	Symptoms	
p1	40	2139	Female	Mild	
p2	45	2545	Male	Severe	
p3	45	2238	Female	High	
p4	38	2843	Male	Severe	
p5	42	2341	Female	Low	
p6	40	2548	Male	Low	
p7	45	2544	Female	A-symptomatic	
p8	32	2538	Male	Low	

Proposed (θ∗, k)-utility: The sensitive partitioning SA ={A1sa,A2sa,A3sa,⋯,Ansa}, and the quasi partitioning QI ={A1qi,A2qi,A3qi,⋯,Anqi}, of the transformed 1:M microdata T into 1:1 microdata linked through QS-Loose linkability, that produces two tables: Sensitive Table (ST) and Quasi Table (QT). The ST consists of SA and BID, where the QT consists of age, zipcode, gender and BID. Below Eq. (5) depicts the proposed approach. (5) iff|∀τiϵT:Ansa←CountDistAisa≤θ|≥2k∧∀τi:Aisa⋅BIDϵsbt∧Aisi⋅BIDϵgenData

where τi represents the tuples from the complete dataset T, having the maximum SA values belonging to CtgT in a transformed 1:1 record shape. So, in the first half of the equation for creating the ST, the proposed approach will execute for checking the θ condition from the θ-Sensitive k-Anonymity approach if the total number of tuples are greater than the user input (i.e., k size, read proposed Algorithm 1 at line 21). The second part of the equation finalizes the sbt and genData (i.e., the ST and QT respectively), and which are the tables obtained through the proposed (θ∗, k)-utility algorithm.

The (θ∗, k)-utility Algorithm

The working of the proposed (θ∗, k)-utility Algorithm 1 is partitioned into three major parts; transformation (lines 3-18), sensitive buckets creation (line 20-27), and quasi generalized buckets creation (lines 29-35). Initially, the 1:M data in T is in its original form, and the sensitive buckets (sbt) and the quasi generalized data (genData) are taken as empty sets. The algorithms begin by computing distinct SAs in T (lines 3-5), which are further categorized into five different sensitive categories to create category CtgT at line 7 (Table 3).

____________________________ Algorithm 1 (θ∗,k)-utility_____________________________________________________________________________________________ Require:     T: 1:M Microdata Table;      k: k-anonymity;      Γ: T populated with CtgT ;      τ: Individual tuple from transform Table (T  );      ℏ: Individual history tuple from history table H; Ensure:     QT: Quasi Table :-genData;      ST: Sensitive Table :-sbt;      ___________________________________________________________________________________________________________________________________________________  1:  sbt={};  2:  genData={};  3:  for all tsai ⋅⋅⋅tsan in T do  4:     Dsan := Compute(Distinct(SA value))  5:  end for  6:  for all dsai ⋅⋅⋅dsan do  7:     CtgT := Categorize Dsan into five categories  8:  end for  9:  for all tsai ⋅⋅⋅tsan do 10:     Γ:= T ↔ CtgTcat 11:  end for 12:  for  all ri ⋅⋅⋅rn do 13:     for all ti ⋅⋅⋅tn do 14:         τi:= max(tsai)∀tsa i  ϵ CtgTcat 15:         ℏi:= ¬max(tsai)∀tsa i  ϵ CtgTcat 16:     end for 17:  end for 18:  T  := ∑n     i=1 τi 19:  Hi:= ∑n     i=1 ℏi 20:  while T     ⁄= {} do 21:     if T     ≤ 2k then 22:         sbtk :=   T 23:         sbt := sbt ∪sbtk 24:     else 25:         Apply θ-Sensitive k-Anonymity Khan, Razaullah and Tao, Xiaofeng and Anjum, Adeel and Kanwal, Tehsin             and Malik, Saif Ur Rehman and Khan, Abid and Rehman, Waheed Ur and Maple, Carsten (2020) 26:     end if 27:  end while 28:  N := —T  qi—   // BID is obtained from bksa 29:  while N ⁄= {} do 30:     if N ≤ 2k then 31:         genData := N 32:     else 33:         genData := genData ∪ gen(N)   // Linked via BID 34:     end if 35:  end while 36:  return  sbt 37:  return  genData____________________________________________________________________________________________________________________________

The CtgT will be used as a reference table while creating sbt. Lines 9-11, populate the original 1:M Table T by assigning the categorical SA values (i.e., Low, Mild, High, Severe, A-symptomatic) to the actual SA (i.e., symptoms attribute), shown in Table 4. The for loop (lines 12-17), creates transform table T; Table 10 at line 14 (i.e., see definition 7). Table 10 has 1:1 microdata. Lines 12 and 13 checks the number of tuples (ti) that belongs to a single individual record (ri), i.e. 1:M data. Line 14 creates the transformed tuples (τi) by selecting high weighted categorical sensitive attribute values. The sensitive vertical attack (sVer) in definition 5 is prevented at this step of the algorithm. By creating the transformed Table 10 from the original Table 1, the leaf-nodes (i.e., in Fig. 2) cannot be correlated with any generalized categorical SA value. Since all the generalized categorical SA values are on the same level in the hierarchy, SA values on a single level cannot be vertically correlated to breach the privacy of an individual. The remaining categorical sensitive values are stored in a history table ℍ (Table 11) at line 19, to avoid any wastage of data.

Next, the algorithm will create sensitive buckets from the transformed data T. The while loop processes all the tuples in T to create k (i.e., user input) size sensitive buckets (sbt). If the tuples to anonymize in T are less than 2 k the algorithm will create final sbt, otherwise it will process sensitive part of all tuples using the θ-Sensitive k-Anonymity algorithm (Khan et al., 2020), to create more diverse Equivalence Classes (ECs) for ST (i.e., Table 12). Since the sbt obtained through the θ-Sensitive k-Anonymity algorithm (Khan et al., 2020), the EC produced can prevent the sva (i.e., Definition 3), and csa (i.e., definition 4). The prevention of data from sva and csa have already been proved in the algorithm of θ-Sensitive k-Anonymity (Khan et al., 2020).

Table 11 History table.

Patient Record ID	Tuple ID	Name	Age	Zipcode	Gender	Symptoms	
p1	t1	Susan	40	2139	Female	Low	
	t2	Susan	40	2139	Female	Mild	
p2	t4	Ronald	45	2545	Male	High	
p3	t6	Keran	45	2238	Female	Mild	
p4	t8	Heather	38	2843	Male	High	
p5	t10	Cytnthia	42	2341	Female	Low	
	t11	Cytnthia	42	234a1	Female	Low	
p6	t14	Peter	40	2548	Male	Low	

Table 12 Anonymized data obtained via proposed ( θ∗, k)-utility algorithm.

(a) Quasi table (QT)	
Patient Record ID	Age	Zipcode	Gender	BID	
p1	(40-42) [40]	(2139-2341) [2319]	Female	1	
p5	(40-42) [42]	(2139-2341) [2341]	Female	3	
p2	(40-45) [45]	(2545-2548) [2545]	Male	1	
p6	(40-45) [40]	(2545-2548) [2548]	Male	4	
p3	(45-45) [45]	(2238-2544) [2238]	Female	2	
p7	(45-45) [45]	(2238-2544) [2544]	Female	3	
p4	(32-38) [38]	(2538-2843) [2843]	Male	2	
p8	(32-38) [32]	(2538-2843) [2538]	Male	4	
n1	(32-38) [32]	(2538-2843) [2538]	Male	4	
(b) Sensitive table (ST)	
Symptoms	BID	
Mild	1	
Severe	
High	2	
Severe	
Low	3	
A-symptomatic	
Low	4	
Low	
Mild	

The last part of the algorithm (line 29-35) creates a generalized QT i.e., Table 12A. The QIs are anonymized at lines 31 and 33 in such a way that the adversary’s confidence about the presence of an individual (in definition 1: ma) or confidence over the absence (i.e., Definition 2-nma) is prevented. The obtained k-anonymized quasi buckets (kb) are linked through the Bucket ID (BID) with the sbt using the one-to-many correspondence (QS-Loose linkability) between the two sub-tables, i.e., Tables 12A and 12B. In Table 12A, the Patient Record ID column is not part of the published table. Finally, the algorithm returns the genData in the form of QT, and sbt in the form of ST, linked through BIDs. The tables obtained from the proposed Algorithm 1 are shown in Tables 12A and 12B. The one-to-many correspondence between QT and ST is the loose linkage (in definition 6) between a single sbt in ST with more than one tuples in different ECs in QT. The EC4 in Table 12A adds a noise tuple (i.e., n1) correspondent to the already added noise SA value in Table 12B because of the θ requirements in θ-Sensitive k-Anonymity algorithm (Khan et al., 2020). The beauty of the QS-loose linkability is the improved utility of the data. Because it allows the least distance QI values to create an EC that can be linked with more than one sbt in ST. Another beauty is the improved privacy implementation. Because the adversary’s confidence to uniquely identify a tuple that belongs to an individual, is reduced by linking a single sbt in ST with more than one k-anonymized ECs in QTs.

HLPN analysis of (θ∗, k)-utility algorithm

The different attacks discussed in Section 5 and 6 are mitigated through the proposed (θ∗, k)-utility algorithm.

The data owner, data publisher, adversary are used to model HLPN for the (θ∗, k)-utility algorithm in Fig. 5. The types showed in Table 13 and data places with description in Table 14. For (θ∗, k)-utility algorithm initially find distinct SAs values. The Rule 6, is used to compute distinct SAs sensitive attributes. In Rule 6 as: (6) R(Compute(Dist(SA))):=∀i2ϵx2,i3ϵx3|i31:=Dist(SA)i21∧x3′:=x3∪i31

Figure 5 HLPN for (θ∗, k)-utility algorithm.

Table 13 Types used in HLPN for ( θ∗, k)-utility.

Symbols	Description	
DSA	Distinct set of sensitive attributes	
SAC	Sensitive attributes consist of 5 categories	
GSA	Generalized SA	
RRT	Remaining Repeated Tuples	
k	Input for ECs	
CF	Condition for k size	
QS	Set of implemented θ ECs	
Anmsa	Non-membership knowledge for sensitive values	
Amasa	Membership knowledge for sensitive values	
τ i	Transform into 1:1 tuples	

Table 14 Mapping of data types in ( θ∗, k)-utility.

Symbols	Description	
ϕ(T)	ℙ (Aid× Aqi× Asa)	
ϕ(CDSA)	ℙ (DSA)	
ϕ(CtgT)	ℙ (SAC)	
ϕ(GenSA)	ℙ (GSA)	
ϕ(History Table)	ℙ(RRT)	
ϕ(Transformed Table)	ℙ(τi)	
ϕ(K-value)	ℙ (k)	
ϕ(Condition found)	ℙ (CF)	
ϕ(DSA)	ℙ (QS)	
ϕ(QT)	ℙ (Aqi× BID)	
ϕ(ST)	ℙ (Asa× BID)	
ϕ(BK)	ℙ(Aid× Aqi× Asa)	
ϕ(SA disc)	ℙ (Aiid× Aiqi× Aisa)	
ϕ(MA disc)	ℙ (Aiid× Aiqi× Amasa)	
ϕ(NMA disc)	ℙ (Aiid× Aiqi× Anmsa)	

After this by using Rule 7, categorized symptoms into five different sensitive categories to create category Table 3. In Rule 7 as: (7) R(Categorize):=∀i4ϵx4,i5ϵx5|i51:=i41∧i52:=i52∧x5′:=x5∪i51,i52

The Rule 8, populate the Table 1 by assigning the categorical SA values (i.e., Low, Mild, High, Severe, A-symptomatic) to the actual SA (i.e., symptoms attribute) in Table 4. In Rule 8 as: (8) R(Populate):=∀i6ϵx6,i7ϵx7|i71:=Populate i61∧x7′:=x7∪i71

Rule 9 is used for selecting a high weighted categorical attributes sensitive value to transformed in Table 10. In the transformation process ignored records stored in History Table 11, it consists of remaining categorical sensitive values, that are stored in a History Table 11 to avoid any wastage of data in Rule 10. The 2 k input take using Rule 11. The k value checked using Rule 12. The threshold value calculated using Rule 13, it will process sensitive part of all tuples using the θ-Sensitive k-Anonymity algorithm (Khan et al., 2020), to create more diverse Equivalence Classes (ECs) for Sensitive Table (ST) Table 12B. (9) R(Max):=∀i8ϵx8,i9ϵx9|i91:=maxi81∧x9′:=x9∪i91

(10) R(Remaining CSA):=∀i10ϵx10,i11ϵx11|i111:=¬maxi101∧x11′:=x11∪111

(11) R(Input K):=∀i12ϵx12,i13ϵx13|i131:=inputi121∧x13′:=x13∪131

(12) R(Check k):=∀i14ϵx14,i15ϵx15|i151:=checki141∧x15′:=x15∪151

(13) R(Applyθ-Sensitive k-Anonymity):=∀i16ϵx16,i17ϵx17|i171:=θ-Sensitive k-Anonymityi161∧x17′:=x17∪171

The splitting is performed for QA and SA attributes, and both are linked with one-to-many correspondence using Rule 14. The obtained k-anonymized quasi buckets are linked through the Bucket ID (BID) with the sbt using the one-to-many correspondence between the two tables. Finally, the algorithm returns the Table 12A QT and Table 12B ST linked through BIDs. In Rule 14 as: (14) R(Splitting):=∀i18ϵx18,i19ϵx19,i20ϵx20|i191:=Spliti181∧i192:=BIDi181∧x19′:=x19∪i191,i192i202:=Spliti181∧i202:=BIDi181∧x20′:=x20∪i201,i202

(15) R(MA Attack):=∀i21ϵx21,∀i22ϵx22,∀i23ϵx23,∀i24ϵx24|MADisi211,i222→i211i221∪i232≠i22∧i23i242∪i243=∅

(16) R(NMA Attack):=∀i25ϵx25,∀i26ϵx26,∀i27ϵx27,∀i28ϵx28−NMADisi251,i272→i282∧NMADisi261,i272=∅

(17) R(SV er Attack):=∀i29ϵx29,∀i30ϵx30,∀i31ϵx31,∀i32ϵx32|SaDisci291,i301→i291,i1301∪i312:=i22∧i23i322∪i323=∅

(18) R(QS − Loose Linkability):=∀i33ϵx34,∀i34ϵx34,∀i35ϵx35,∀i36ϵx36|i331,i341→i331,i1341∪i352:=i22∧i23i362∪i363=∅

(19) R(SV A):=∀i39ϵ,x39,∀i40ϵx40,∀i42ϵx42,∀i43ϵx43|SaDisci392,i402,i422≠i21∪i22∪i23i432∪i433=∅

In Rules 15 and 16, the adversary’s confidence about the presence of an individual, or confidence over the absence is prevented. The sensitive vertical attack (sVer) prevented by creating the Table 4 from the Table 1, so SA values on a single level cannot be vertically correlated to breach the privacy of an individual in Rule 17. The adversary’s confidence to uniquely identify a tuple which belongs to an individual, is reduced by linking a single sbt in ST with more than one k-anonymized EC in QTs in Rule 18. The diverse EC produced, can prevent the sva and csa. The prevention of data from sva and csa in Rule 19. The (θ∗, k)-utility algorithm protects from above mentioned attacks, results in form of a null value as shown in Rules 15, 16, 17, 18, and 19.

Experimental Analysis

In this section, we analyse the comparative results of our proposed (θ∗, k)-utility technique for 1:M microdata, in terms of utility, privacy, and computational efficiency.

The anonymized data quality and the execution time is measured to compare proposed (θ∗, k)-utility algorithm, with θ-Sensitive k-Anonymity (Khan et al., 2020), (k, l)-diversity (Gong et al., 2017), and (p, l)-angelization (Kanwal et al., 2019), algorithms.

Experimental setup

All the experiments are performed on a machine having Windows 10 operating system with Core i7 processor and 8GB RAM. The proposed algorithm is implemented in Python 3.9 language. We used the modified ‘Adult’ dataset, which is publically accessible from the repository of UC Irvine machine learning.1 ,2

In the modified Adult dataset the age, zipcode, and gender are considered as QIs, while the symptoms attribute is considered as the SA.

The anonymized data obtained from the proposed and the base algorithms are analyzed for utility using normalized certainty penalty (NCP) (Anjum et al., 2018) and query accuracy (Anjum et al., 2018), for privacy using the average number of vulnerable records, and the computational efficiency is analyzed with the average execution time of all the algorithms.

Utility loss

The utility loss of the anonymized data is measured using the following techniques.

Normalized certainty penalty

Normalized certainty penalty (NCP) is one of the techniques which measures the utility loss caused by data anonymization. We measure the utility loss caused by the QIs. High penalties indicate high utility loss and vice versa.

Let T ={q1,q2, …,qm} are QI. The utility loss for a single QI attribute is shown in Eq. (20) as. (20) NCPqit=xi−yiQi

where yi ≤ zi ≤ xi, and zi is the actual QI value from T, and |Qi—is the domain range on QIi, i.e., max{t.QIi} − min{t.QIi}. The total weighted certainty penalty for the whole table is the sum of all attributes in a tuple and then adding NCP obtained from all tuples, as in Eq. (21). (21) NCPT∗= ∑t=T∗∑i=1qwi⋅NCPqit

where, NCP(t) = ∑i=1qwi⋅NCPqi (t) represents penalty for a tuple, wi are weights associated to attributes, and T∗ is the final anonymized release.

Figure 6 shows the NCP for utility measurement on anonymized release. Figure 6 shows the comparative results of θ-Sensitive k-Anonymity, (k, l)-diversity, (p, l)-angelization and the proposed (θ∗, k)-utility algorithms, with varying k for the complete dataset. The increase in the graph values shows more utility loss. The higher value of k collectively for all algorithms results in higher utility loss because of the increased generalization range in each EC. For (k, l)-diversity, it is impossible to satisfy both k-anonymity and l-diversity constraints at the same time to achieve high privacy with minimum information loss, where the high value of the l-diversity is not recommended for high value of k. However, still the (k, l)-diversity results shows better utility than the θ-Sensitive k-Anonymity and the (p, l)-angelization algorithms, because the (p, l)-angelization algorithm only extends the (k, l)-diversity (which works only for 1:1 dataset) for the 1:M-MSA data without considering utility and privacy of the anonymized data. In (p, l)-angelization, increasing the diversity in for sensitive attributes in SAFBs greatly reduces the utility of the k-anonymous groups because of the one-to-one correspondence between the ST and QT. Through the one-to-one correspondence between the ST and QT, any change in ST directly affect the QT for the same changes. So with respect to utility (p, l)-angelization is more worse than (k, l)-diversity. The multiplicative increase in utility loss for θ-Sensitive k-Anonymity is because of its straight forward privacy implementation for single SA without any contribution for utility improvement in the developed algorithm.

Figure 6 Normalized certainty penalty.

Our proposed (θ∗, k)-utility algorithm, independently generalizes the QI values to create less distance ECs for any size of k, which results in low utility loss as compared to its counterparts. Here, the k size in QT is not affected by any changes in ST. The proposed (θ∗, k)-utility algorithm in Fig. 6B depicts the NCP with varying size of data set and for a fixed value of k =4. The utility loss reduces with more and more 1:M records is because of the availability of more suitable QI values from the increased dataset to create smaller distance ECs, which reduces the loss in data utility. Our proposed (θ∗, k)-utility algorithm produces better results as compared to its counterparts, and depicts almost a constant data utility with any number of records. This is because of the separate publishing of the QT from the ST, which is enabled by the one-to-many correspondence approach.

Query accuracy

Query accuracy measures the utility loss between the original and anonymized release using an aggregate query, e.g. COUNT, AVG, SUM etc. Consider the following aggregate query in Eq. (22). (22) SELECTCOUNT∗fromT∗whereA1qiϵDomainA1qiAND⋯ANDAmqiϵDomainAmqi

Anonymized table T* has m as a total Aqis, i.e., Aiqi,Aiqi, …,Aqqi. The domain size i.e., (Aiqi) depends on query selectivity (θ) which indicates the expected number of tuples selection from an executed aggregate query. The tuple selectivity can be seen in in Eq. (23). (23) θ=tqT

where |T| is the total number of tuples in the dataset and |tq| indicates the number of tuples obtained from a query (Q). To measure the utility loss, the query error in Eq. (24) analyzes the error between the COUNT queries executed on the published and original dataset. (24) QueryError=|Countanonymized−Countoriginal|/Countoriginal

Query error is a common matrix to measure the utility of the anonymized release. We perform utility loss analysis between the θ-Sensitive k-Anonymity, (k, l)-diversity, (p, l)-angelization, and the proposed algorithm of (θ∗, k)-utility, by generating 1000 randomly queries and averaging their query error in Fig. 7.

Figure 7 Query error.

Figure 7A shows the query error for varying k size. For all the comparative graphs, with increase in k size the query error increases, because high k means high distance ECs. Then the aggregate query results in more number tuples on anonymized data as compared to the original data. So the high comparative difference between the original and anonymized data results in an increased query error rate. The (p, l)-angelization do no focus on the utility of the data at all because even the separately created sensitive and quasi tables are not considered as actual separate tables because they are still directly connected through the one-to-one buckets in both tables. While in (k, l)-diversity although both QI and SA are generalized separately, even though, both are dependent on k, for measuring the utility of any k-anonymous group. The values obtained from the θ-Sensitive k-Anonymity indicates its better utility than the (p, l)-angelization and (k, l)-diversity is because of its local generalization. The lowest utility loss by our proposed approach is because of the autonomous ECs of QIs.

Figure 7B shows the query error with respect to varying selectivity for k = 4 and dataset = 6,000. With increase in selectivity (i.e., high predicates) less number of records will be selected, which results in a low error rate in the anonymized data. Again, our proposed (θ∗, k)-utility algorithm has continuously low query error as compared to the θ-Sensitive k-Anonymity, (p, l)-angelization, and (k, l)-diversity techniques because of the independent anonymization of the QT with respect to the ST. The low query error in our proposed approach depicts the low difference in the tuples selection between the original and anonymized releases.

Privacy loss

The privacy loss means re-identification of an individual record in the anonymized dataset. In this work, we are using two different methods to measure privacy loss. One is the actual record intersection method, while another is the probability method.

Record intersection

Loss in the privacy of data is the identification of vulnerable records in the anonymized data which ultimately re-identify an individual record. Equation (25) measures an average number of vulnerable records.

The original dataset contains an input file that contains the total data that is not yet anonymized i.e., Table 1, the output table indicates the published tables in the anonymized form, i.e., QT12a and ST12b. (25) VulnerableRecords=Actual∩Output.

The vulnerability of number of records in (k, l)-diversity and(p, l)-angelization is higher than the θ-Sensitive k-Anonymity and (θ∗, k)-utility as shown in Fig. 8A. This is due to the fact that the (k, l)-diversity uses a transaction generalization technique (Gong et al., 2017). The lowest common cut on transaction generalization hierarchy leaves the single SA from different sub-set unprotected and vulnerable, as shown in Fig. 2 that causes sVer attack which breaches the privacy of that specific individual. In (p, l)-angelization the vulnerability exists due to the sensitive attributes fingerprint correlation attack, as mentioned in Khan et al. (2020c). As in (p, l)-angelization, the QT and ST tables have one-to-one correspondence (discussed in Motivation Scenario III), so the adversary can correlate both tables and can easily create a single table. In that way, the ma and nma attacks (see definitions in Section 3) can be performed on the dataset. The variance-based privacy implementation by θ-Sensitive k-Anonymity is stronger. However for the 1:M dataset, it becomes useless to achieve privacy because in such data each record consists of more than one tuple, and attackers can perform sva, and csa attacks on the dataset. However, the proposed (θ∗, k)-utility further improves its privacy by categorizing the SA into categorical SA values (i.e., Fig. 2), and prevents all such attacks (i.e., ma, nma, sva, csa, and sVer). This reduces the re-identification of an individual record and provides more data protection as compared to its counterparts, as shown in Fig. 8A.

Figure 8 (A) Record intersection; (B) record linkability.

Record linkability

In this subsection, the impact of the privacy parameter record linkability (RL) is analyzed through our proposed (θ∗, k)-utility algorithm in comparison to the (k, l)-diversity, (p, l)-angelization and θ-Sensitive k-Anonymity. RL is a measure of disclosure risk (i.e., privacy loss) and is the probability of correctly linked records between the original and the anonymized data. For a record ti ϵ T, record linkage probability in anonymized form PRLti∗, is calculated using Eq. (26). (26) PRLti∗=1∣ECj∣:tiϵECj0:otherwise

where ECj is generalized group of records in QT with minimum distance from ti. The RL for complete microdata T is then calculated in Eq. (27) as below. (27) RL= ∑tiϵTPRLti∗.

For example, record ti ϵ T is put into EC1 after anonymization. Now for each original record ti find the closest EC in anonymized QT, let say it is EC2. If EC1 is EC2, the record ti is linked and is computed via Eq. (26). We finally sum RL of all original records as shown in Eq. (27). Figure 8B shows the privacy loss (i.e., RL) for the proposed (θ∗, k)-utility, in comparison to the (k, l)-diversity, (p, l)-angelization, and θ-Sensitive k-Anonymity algorithms. The lower value of RL shows the lower privacy loss and vice versa. The probability to link a record from the microdata T is high with an anonymized EC of small size, because the intruder already knows the QIs of the intended individual, and a record can easily be linked with a few number of records group. So the privacy loss with respect to RL for small k is high as compared to high value of k.

Figure 8B shows that the highest privacy loss in (k, l)-diversity, is due to the lowest common cut on the sensitive values (i.e., sVer attack) which becomes vulnerable because of the remaining sub-set of sensitive values unprotected (see Fig. 2). In (p, l)-angelization, the fingerprint correlation attack and the one-to-one correspondence between the QT and ST are the due reasons of privacy losses. In θ-Sensitive k-Anonymity the attackers can easily perform sva and csa attacks because of its inapplicability for 1:M type of data. Figure 8B depicts that the RL in all these approaches is high for small value of k, because the probability of linking the target quasi identifier values with small number of records is high as compared to the larger size EC. However, our proposed (θ∗, k)-utility algorithm not only categorizes the SA values to implement privacy in sensitive values but also contributes in the form of QS-loose linkability for implementing privacy both in QT and ST, and implementing utility in the QT only. The QS-loose linkability not only minimizes the chances of record linkability but almost vanishes it. So, the beauty in the novelty of the proposed (θ∗, k)-utility algorithm is the contribution both for privacy and utility at the same time.

Execution time

Computational efficiency is the overall execution time of an algorithm. Figure 9 shows the execution time of our proposed (θ∗, k)-utility algorithm along with its counterpart techniques. In both Figures i.e., Figs. 9A and 9B, the (p, l)-angelization has the highest execution time because of the weight calculation and handling 1:M-MSA data. The proposed (θ∗, k)-utility algorithm has higher execution time than θ-Sensitive k-Anonymity, because of the additional work of categorising the SA and creating one-to-many loose linkability between QT and ST, along with the variance calculations for SAs. The (k, l)-diversity has the lowest execution time because of the simple approach of the algorithm i.e., only 1:M generalization and splitting the attributes.

Figure 9 Execution time.

Discussion

The results show that the proposed (θ∗, k)-utility algorithm outperforms all its compared counterparts concerning utility and privacy. Our proposed (θ∗, k)-utility algorithm for measuring NCP, independently generalizes the QI values to create less distance ECS for any size of k, which results in low utility loss compared to its counterparts. For the query accuracy, the lowest utility loss by our proposed approach is because of the autonomous ECs on QIs. The (θ∗, k)-utility improves privacy by categorizing the SA into categorical SA values also using the variance-based privacy implementation. The proposed (θ∗, k)-utility algorithm has higher execution time than θ-Sensitive k-Anonymity and (k, l)-diversity because of the additional work along with the variance calculations and has lower execution time than (p, l)-angelization. So our proposed (θ∗, k)-utility algorithm is best to achieve higher data privacy and data utility as compared to its counterparts.

Conclusion and Future work

This article addresses the problem of anonymizing the 1:M microdata with significantly improving the utility of anonymized release. We proposed an anonymization algorithm which prevent any possible attack e.g., membership attack (ma), non-membership attack (nma), sensitive variance attack (sva), categorical similarity attack (csa), and sensitive vertical attack (sVer), which may exists in either θ-Sensitive k-Anonymity (Khan et al., 2020), or in (k, l)-diversity (Gong et al., 2017), or in (p, l)-angelization (Kanwal et al., 2019) techniques. The proposed solution; (θ∗, k)-utility, extends the applicability of θ-Sensitive k-Anonymity (Khan et al., 2020), for anonymizing the 1:M microdata. The (θ∗, k)-utility algorithm executes by taking three proactive steps: transformation, sensitive bucket creation, and quasi generalized buckets creation. The SA values i.e., COVID symptoms, in ST are categorized into Table 2 CtgT i.e., Low, Mild, High, Severe, and A-symptomatic, for the purpose to implement privacy in the Table ST Table 12B. The QS-loose linkability between the Table 12A QT and Table 12B ST, not only implements privacy in both tables but also significantly improves the utility of the anonymized data. The results from experiments depicts that with respect to both utility and privacy the proposed (θ∗, k)-utility algorithm outperforms all its compared counterparts.

For future work considerations, the proposed algorithm can be extended to implement privacy in a dynamic data publishing scenario (Xiao & Tao, 2007; Khan et al., 2020b) for periodic or non-periodic updates. Similarly, the proposed work can be extended to a cluster based anonymization technique to more efficiently overcome the problem of privacy and utility paradigm (Safi & Hwang, 2022). Another privacy extension can be privacy-preserving federated learning (PPFL) (Yin, Zhu & Hu, 2021). PPFL is a collaborative training process based on iterative model averaging where the user generated data is not directly shared with any third party which greatly benefits the used data from not being disclosed to any un-identified intruder.

Supplemental Information

Supplemental Information 1 Code in python

Click here for additional data file.

Additional Information and Declarations

Competing Interests

Author Contributions

Data Availability

1 DOI: 10.5281/zenodo.7214275, also in original can be seen on: https://archive.ics.uci.edu/ml/datasets

2 Some examples of modified datasets were used in other research as a reference. https://archive.ics.uci.edu/ml/datasets, https://datahub.io/machine-learning/adult#readme, https://www.researchgate.net/figure/Examples-of-generated-counterfactuals-on-the-modified-Adult-dataset-Example-Based-CF-and_tbl2_337830079, https://openreview.net/forum?id=bYi_2708mKK

The authors declare there are no competing interests.

Rabeeha Fazal conceived and designed the experiments, performed the experiments, performed the computation work, prepared figures and/or tables, authored or reviewed drafts of the article, and approved the final draft.

Razaullah Khan conceived and designed the experiments, performed the experiments, performed the computation work, prepared figures and/or tables, authored or reviewed drafts of the article, and approved the final draft.

Adeel Anjum conceived and designed the experiments, analyzed the data, performed the computation work, authored or reviewed drafts of the article, and approved the final draft.

Madiha Haider Syed conceived and designed the experiments, analyzed the data, authored or reviewed drafts of the article, and approved the final draft.

Abid Khan conceived and designed the experiments, analyzed the data, authored or reviewed drafts of the article, and approved the final draft.

Semeen Rehman conceived and designed the experiments, analyzed the data, authored or reviewed drafts of the article, and approved the final draft.

The following information was supplied regarding data availability:

The data set used in the work is derived from the publicly available dataset available at the UC Irvine Machine Learning Repository: https://archive.ics.uci.edu/ml/datasets/adult.

The derived data set is available at Zenodo: Ronny Kohavi, & Barry Becker. (1996). UCI Machine Learning- Adult Dataset [Data set]. Zenodo. https://doi.org/10.5281/zenodo.7214275.

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
