# Peer review of "Improved angelization technique against background knowledge attack for 1:M microdata"

_PeerJ Computer Science, doi:10.7717/peerj-cs.1255_

## Round 0.1 · original submission · Major Revisions

Please address all comments and submit a response letter detailing all corrections made.

Reviewer 1 ·

Basic reporting

No comment.

Experimental design

The proposed proposed (θ*, k)-utility algorithm, independently generalizes the QI values to create less distance ECs for any size of k, which results in low utility loss as compared to its counterparts. However, the authors need to better explain the contribution of this algorithm except of the transformed 1:M microdata T into 1:1 microdata. They also need to focus on the equation (5). How the input value (2k) is selected to θ -Sensitive k-Anonymity algorithm?

Last, it is known that the θ-Sensitivity, is the product of variance (σ^2) and
Observation 1 (μ). Τhe symbolism (θ*) in the proposed algorithm indicates some optimization in the parameter (θ)? Τhis symbolism needs clarification.

Validity of the findings

The experimental results of the proposed (θ*, k)-utility technique for 1:M
dataset outforms in terms of utility, privacy, and computational efficiency. Ηowever some clarifications are needed:
- In Figure 8, the privacy loss is zero for all values (k)? This needs clarification.

Additional comments

The section about the "Future Work" must be extended. For example, would it be helpful to use cluster-based anonymization algorithms to construct the hierarchical structure of the sensitive attributes of Figure 2?

Reviewer 2 ·

Basic reporting

The paper is well structured. The analysis of the findings is very interesting. Literature reference is relevant and covers the topics but the main contributions of the paper are not sufficient to justify another paper on top of what already exists in the literature

Experimental design

As a concept, it is well designed. The proposed scheme has certainly been used in the right way but simulations results presented in this paper need more work to from being comprehensive.

Validity of the findings

There is not much novelty and contribution in the performance evaluation.

Additional comments

The subject of this paper is in the right direction on a very important issue of privacy and anonymized data, but more effort is needed to become competitive. The main technical drawback is the lack of originality and novelty in the paper.

·

Basic reporting

This paper addresses the problem of anonymizing the 1:M microdata by significantly improving the
utility of anonymized release. The manuscript has merit, but the following comments should be answered carefully should I recommend publication of this work.
1. A detailed description must be included in the paper that emphasizes the main pros and cons of the authors’ proposal with regard to the state of the art.
2. The authors must extend the explanation of how the proposed technique can be extended to cover a wider scientific area without reducing the main points that are currently described.
3. I think that if the authors wish this paper is well considered by experts in the cyber security communities, more attention should be devoted to discussing the application scenario. I suggest simplifying it or better explain with realistic examples.

Experimental design

The authors should probably provide more information about the proposed architecture. This is a major issue of the paper of how the authors have chosen this specific architecture for the proposed processing method, how it emerged and why the proposed architecture is the optimal solution.

Validity of the findings

A major issue with the paper is the explanation of the results, which are presented casually and without thorough analysis.

Additional comments

The figures are small and apparently of low resolution. If the authors consider that it provides important information, they should definitely enlarge it to be clear and legible.

---

## Round 0.2 · Minor Revisions

Please revise the article in order to make the originality clear to the readers.

Reviewer 1 ·

Basic reporting

Τhe authors improved the paper considerably and responded satisfactorily to the reviewers' comments.

Experimental design

No comment

Validity of the findings

No comment

Additional comments

Τhe authors improved the paper considerably and responded satisfactorily to the reviewers' comments.

Reviewer 2 ·

Basic reporting

The paper is well structured. The analysis of the findings is very interesting. Literature reference is relevant and covers the topic

Experimental design

The proposed scheme has certainly been used in the right way and the authors in the revised manuscript, try to clarify the experiment results.

Validity of the findings

In the performance evaluation, the authors need more effort to achieve novelty.

Additional comments

The main technical drawback in the revised manuscript even now is the lack of originality and novelty.

---

## Round 0.3 · accepted · Accept

The article is ready for publication since the authors have completed all needed corrections.